# HIV-1 Tat Protein and Cigarette Smoke Mediated ADAM17 Upregulation Can Lead to Impaired Mucociliary Clearance

**DOI:** 10.3390/cells13232009

**Published:** 2024-12-05

**Authors:** Kingshuk Panda, Maria J. Santiago, Md. Sohanur Rahman, Suvankar Ghorai, Stephen M. Black, Irfan Rahman, Hoshang J. Unwalla, Srinivasan Chinnapaiyan

**Affiliations:** 1Department of Cellular and Molecular Medicine, Herbert Wertheim College of Medicine, Florida International University, 11200 SW 8th Street, Miami, FL 33199, USA; kpand014@fiu.edu (K.P.); msant206@fiu.edu (M.J.S.); mdsrahma@fiu.edu (M.S.R.); sghorai@fiu.edu (S.G.); stblack@fiu.edu (S.M.B.); 2Department of Chemistry and Biochemistry, Florida International University, 11200 SW 8th Street, Miami, FL 33199, USA; 3Center for Translational Science, Florida International University, 11350 SW Village Parkway, Port St. Lucie, FL 34987, USA; 4Department of Environmental Medicine, University of Rochester Medical Center, 601 Elmwood Ave, Rochester, NY 14642, USA; irfan_rahman@urmc.rochester.edu

**Keywords:** HIV-1, cigarette smoke, COPD, *SIRT1*, *ADAM17*, mucociliary clearance

## Abstract

Human immunodeficiency virus type-1 (HIV-1) associated comorbidities account for the majority of poor health outcomes in people living with HIV (PLWH) in the era of antiretroviral therapy. Lung-related comorbidities such as chronic obstructive pulmonary disease (COPD) and bacterial pneumonia are primarily responsible for increased morbidity and mortality in PLWH, even when compensated for smoking. Smokers and COPD patients demonstrate cilia shortening, attenuated ciliary beat frequency (CBF), dysfunctional ciliated cells along with goblet cell hyperplasia, and mucus hypersecretion. This is exacerbated by the fact that almost 60% of PLWH smoke tobacco, which can exacerbate inflammation and mucociliary clearance (MCC) dysfunction. This study shows that HIV Tat alters the microRNAome in airway epithelial cells and upregulates miR-34a-5p with consequent suppression of its target, Sirtuin 1 (*SIRT1*). *SIRT1* is known to suppress Metalloproteinase 17 (*ADAM17*), a protease activating Notch signaling. HIV and cigarette smoke (CS) upregulate *ADAM17*. *ADAM17* upregulation followed by *SIRT1* suppression can lead to decreased ciliation, mucus hypersecretion, and attenuated MCC, a hallmark of chronic bronchitis in smokers and COPD. It is, therefore, essential to understand the pathophysiological mechanism resulting in acquired Notch dysregulation and its downstream impact on HIV-infected smokers.

## 1. Introduction

Human immunodeficiency virus type-1 (HIV-1) infection has been associated with a spectrum of respiratory complications, including chronic obstructive pulmonary disease (COPD) and increased susceptibility to respiratory infection [1]. HIV-associated comorbidities are primarily responsible for the poor quality of life in people living with HIV (PLWH). The FDA has approved 48 medications across 10 classes of antiretroviral drugs for HIV treatment. Most individuals newly diagnosed with HIV initiate therapy with a regimen consisting of either three drugs or a combination of two drugs, typically selected from different antiretroviral classes to enhance efficacy and prevent resistance [2]. Although cART successfully suppresses viral replication, it is unable to eradicate HIV due to the reactivation of HIV from latently infected anatomical reservoirs [3,4,5].

Lung-related comorbidities like COPD and bacterial pneumonia are primarily responsible for increased morbidity and mortality in PLWH, even when compensated for smoking [6]. Patients with HIV die of non-AIDS comorbidities almost a decade earlier than those without HIV [7]. In parallel, cigarette smoke represents a major environmental risk factor for respiratory diseases, and its association with mucociliary dysfunction has been extensively documented. Cigarette smoke contains a complex mixture of toxicants that can directly affect the respiratory epithelium, impairing ciliary function and disrupting mucus production [8]. Mucociliary clearance (MCC) is a critical defense mechanism in maintaining airway homeostasis and protecting the lungs against inhaled pathogens and particles of cystic fibrosis [9,10]. MCC dysfunction is primarily responsible for microbial colonization of airways in COPD and cystic fibrosis. For effective mucociliary clearance (MCC), mucus, cilia, and a thin layer of airway surface liquid (ASL) is crucial to support ciliary beating. Any irregularities in these elements can impair MCC, which can encourage microbial colonization and chronic inflammation.

In earlier studies, we and others have reported that bronchial epithelial cells express canonical HIV receptors and can be infected by the virus [11,12]. HIV Tat and CS increase TGF-β signaling in the airway epithelial cells [13], and the HIV or HIV Tat protein effects can be exacerbated by CS, explaining the increased severity of COPD in PLWH smokers [14,15]. Although significant progress has been made in understanding the individual effects of HIV-1 and smoking on pulmonary complications in comorbidities, the precise molecular mechanism remains incompletely characterized. This is of significant concern since 60% of PLWH are smokers. While HIV Tat and CS have individually been linked to mucociliary dysfunction, the potential synergistic effects of these factors on bronchial airway epithelial cells remain poorly understood. We have shown that HIV Tat and CS upregulate TGF-β signaling, and Tat alters the microRNAome of primary normal human bronchial epithelial (NHBE) cells.

TGF-β signaling plays a vital role in the progression of chronic airway diseases like COPD and lung infection [16,17,18,19]. Moreover, we have shown that TGF-β1 increases the viral burden in the airway [20]. Of note, our studies demonstrated that HIV and CS suppress cystic fibrosis transmembrane conductance regulator (*CFTR*), individually or synergistically, through TGF-β1 signaling, which plays a major role in MCC [11,15]. Smokers and COPD patients demonstrate cilia shortening, attenuated CBF [21,22], dysfunctional ciliated cells [23,24], goblet cell hyperplasia, and mucus hypersecretion. Several studies have shown that HIV Tat suppresses *SIRT1* [25]. *SIRT1* has been shown to suppress *ADAM17*, a protease involved in activating Notch signaling [26], as well as IL-6 receptor (IL-6R) and TNF-α [27]. Notch signaling upregulation dysregulates cell fate determination, leading to goblet cell hyperplasia and ciliary cell metaplasia observed in chronic airway diseases like asthma and COPD. Hence, *SIRT1* suppression can depress MCC, leading to recurrent lung infections and inflammation, and it is important to understand the pathophysiological mechanisms that lead to acquired Notch dysregulation. The novelty of this study is to investigate *SIRT1* and *ADAM17*’s role in lung inflammation exacerbating mucociliary dysfunction in NHBE cells in the context of HIV and CS. By employing state-of-the-art experimental approaches, this study seeks to contribute valuable insights into the complex interaction between *SIRT1* and *ADAM17* that may underlie respiratory complications in individuals living with HIV, particularly those with a history of CS exposure. The findings from this investigation may lead to targeted interventions aimed at preserving mucociliary clearance and inflammation, and improving respiratory health in vulnerable populations.

## 2. Materials and Methods

### 2.1. Cell Culture

Primary human bronchial epithelial (NHBE) cells were isolated from lungs procured through the University of Miami Life Alliance Organ Recovery Agency (LAORA). These lungs, which were not suitable for transplantation and were made available for research by the donors’ families, were obtained with limited demographic information, including age, sex, smoking history, and any history of infections such as HIV, HCV, EBV, or CMV, as well as COPD status and pack-years for smoker’s lungs. Although the material contains minor, de-identified information relating to deceased individuals, its use is not an investigation of human subjects as defined in CFR 46.102. For lungs donated solely for research purposes, the LAORA maintains everyone’s signed consent (or legal healthcare proxy). The NHBE cells were re-differentiated at the air–liquid interface (ALI) as described by our laboratory and Fulcher and Randall [15,20,28,29,30], during which it underwent differentiation and produced both the in vivo morphology and key physiologic processes to regenerate the native bronchial epithelium ex vivo [29,30]. The experiments were conducted using NHBE-ALI cultures from non-smokers to avoid any interference with the results. In addition, we used BEAS-2B cells, which is a transformed human bronchial epithelial cell line obtained from CRL-9609 of the American Type Culture Collection (ATCC) in Manassas, VA, USA. The cells were grown in a BEGM growth medium as per the supplier’s instructions for the microRNA (miRNA) mimic experiment. All cell cultures were incubated in a humidified environment at 37 °C and a 5% CO_2_ atmosphere.

### 2.2. CS Exposure

The NHBE ALI cultures were exposed to CS using a SCIREQ smoke robot (Montreal, QC, Canada). They used four research-grade cigarettes (Research cigarette, University of Kentucky, College of Agriculture, Cigarette Program, code: 3R4F) with a puff volume of 35 mL for 2 s every 60 s. The smoke was then blown over a cell culture filter at a rate of 5 mL/min, as per ISO 3308 [31]. The total number of puffs was 32, which took approximately 35 min. Smoke exposures were done 24 h before infecting the cells with R5-tropic strain HIV. CS was applied every three days for chronic smoke exposure using the same regimen.

### 2.3. Infection Studies and Proteins

NHBE ALI cultures were grown on trans wells and infected with the R5-tropic viral strain (HIV BaL) at a 5ng/mL p24 equivalent concentration. As described previously, the infection was carried out on both apical and basolateral sides [11,20]. Virus residue was removed from both sides of the cells by washing four times with PBS after 16 h post-infection. To confirm that all input viruses had been removed, the last PBS was collected for p24 analysis. Active infection was confirmed by analyzing p24 levels in a culture medium on Day 3 using a p24 ELISA kit (ZeptoMetrix Corp., NY, USA, Cat # 0801200) according to the manufacturer’s instructions. For chronic HIV exposure, infection was allowed to proceed for up to 9 days. The HIV-1 IIIB Tat recombinant protein (Catalog #HRP-2222) was obtained through the NIH HIV Reagent program. It was reconstituted in PBS containing 1 mg/mL BSA and 0.1 mM DTT to prepare a stock solution. Recombinant TGF-β1 (R&D Systems, MN, USA, Cat #240-B-002) was dissolved according to the manufacturer’s instructions at a 10 µg/µL stock concentration. The final concentration for all treatments was 10 ng/mL.

### 2.4. microRNA Mimic and siRNA Transfection in BEAS-2B Cells

BEAS-2B cells were seeded on collagen-coated 12-well tissue culture plates and cultured for 24 h up to 70% confluence. Then, the cells were transfected with a 40 nM mimic of has-miR-34a-5p (ThermoFisher Scientific, MA, USA, Cat # 4464066) using lipofectamine RNAiMAX (ThermoFisher Scientific/Invitrogen, MA, USA, Cat # 13778075) with Opti-MEM I reduced-serum medium (Gibco, NY, USA, Cat # 31985070). Another experimental set of cells was transfected with 40 nM siSIRT1 (ThermoFisher Scientific, Cat # AM16708) using lipofectamine RNAiMAX with Opti-MEM medium. Separately, cells were treated with lipofectamine RNAiMAX in an Opti-MEM medium that was used as a control. After 48 h RNA and protein samples were collected from cells and were lysed using the respective lysis buffer.

### 2.5. Quantification of mRNA Expression by qPCR

Total RNA was isolated from treated or infected or smoked cells using PureLink RNA mini kit (ThermoFisher Scientific/Invitrogen, MA, USA, Cat # 12183020). The RNA purity and concentration were measured using a microplate RNA reader (Synergy HT Multi-Mode Microplate Reader from BioTek, Winooski, VT, USA). The high-capacity cDNA reverse transcription kit (Applied Biosystems, MA, USA, Cat# 4368814) was used to prepare the cDNA as per the manufacturer’s protocol. The mRNA expression levels were measured by Bio-Rad CFX96 real-time system and used different TaqMan probes (Life Technologies/Applied Biosystems, MA, USA; hsa-miR-34a-5p, Cat # 4427975; *SIRT1*, Cat # Hs01009006_m1; *ADAM17*, Cat # Hs01041915_m1; *MUC5AC*, Cat # Hs01365616_m1; *GAPDH*, Cat # Hs02758991_g1) in combination with TaqMan^TM^ fast advanced master mix (Life Technologies/Applied Biosystems, Cat # 4444557) according to the manufacturer’s directions. The results were obtained as a relative quantification normalized against an internal control (GAPDH).

### 2.6. Western Blot Analysis

Total protein was isolated from treated or infected or smoked cells and was lysed with an RIPA (radioimmunoprecipitation assay) buffer (ThermoFisher Scientific, Cat # 89901) mixed with a protease inhibitor cocktail (ThermoFisher Scientific, Cat# 78429), followed by sonication. The protein concentration was quantified by a nanodrop spectrometer using the BSA method (ThermoFisher Scientific, USA). Equal amounts of total protein were loaded on 4–20% precast polyacrylamide gel (Bio-Rad, CA, USA, Cat # 4568094) run at 100 V and transferred onto a polyvinylidene difluoride (PVDF) membrane. The PVDF membrane was blocked with a 10% blocking solution for 1 h. Afterward decant the blocking solution was decanted and primary antibodies like SIRT1 (1:1000 dilution; Cell signaling, MA, USA, Cat # 9475S), ADAM17 (1:1000 dilution; Cell signaling, Cat # 61048S), α-tubulin (1:2500 dilution; Cell signaling, Cat # 2125S), and GAPDH (1:2500 dilution; Cell signaling, Cat # 2118S) were added, with an overnight incubation mixed with 5% blocking solution. The blot was then washed with TBS (Tris-buffered saline, Bio-Rad, Cat # 1706435) mixed with Tween 20 (Sigma-Aldrich, MA, USA, Cat # 9005645) and incubated for 1 h with horseradish-peroxidase-conjugated anti-rabbit (Promega, WI, USA, Cat # W4011) or anti-mouse (Promega, Cat # W4021) secondary antibody (1:2500 dilution) mixed with 1% blocking solution. As described by the manufacturer, the protein bands were detected in Chemidoc (Bio-Rad Laboratories, USA) using the super signal West femto maximum sensitivity substrate (ThermoFisher Scientific, Cat #34095). The density values were calculated using quantity Image Lab Touch Software (Ver. 3.0.1.14) (Bio-Rad Laboratories, USA) and normalized to α-tubulin and GAPDH.

### 2.7. Immunofluorescence Imaging Analysis

CS and HIV-infected NHBE ALI cultures were rinsed using DPBS (Gibco, Cat # 14040-133) and fixed by adding 4% (*w*/*v*) paraformaldehyde diluted in PBS (ThermoFisher Scientific, Cat # J19943-K2) into the basal and apical sides, followed by incubation for 15 min at room temperature (RT). Then, cells were washed 3 times in PBS for 10-min intervals, and membranes were excised from the insert and submerged with 1% Triton-X-100 (Sigma-Aldrich, Cat # T8787) for 15 min at RT. Next, the non-specific binding sites were blocked using 3% BSA (Roche, IN, USA, Cat # 9048468) in PBS and permeabilized cells for 1 h at RT. Then, the cells were washed 3 times in PBS for 10-min intervals and specific primary antibodies were added: MUC5AC staining for anti-mucin MUC5AC (Sigma-Aldrich, Cat # MAB2011), and cilia staining for anti-acetyl-alpha-tubulin (Lys40) (Sigma-Aldrich, Cat # SAB5600134) incubated overnight at 4 °C. The primary antibody was removed and washed 3 times in PBS for 10-min intervals. After washing, the membranes were incubated for 1 h at RT with their respective secondary antibodies conjugated to fluors (Donkey anti-Mouse IgG Alexa fluorTM 488, Invitrogen, Cat # A2120, and Chicken anti-Rabbit IgG Alexa fluor^TM^ 488, Invitrogen, Cat # A21441). Afterward, the membrane was washed with PBS and mounted with 4′, 6-diamidino-2-phenylindole (DAPI) containing Fluoromount-G^®^ (SouthernBiotech, AL, USA, Cat # 0100-20) for nuclear staining. Immunofluorescent images were acquired using BZX700 Keyence ALL in one microscope and analyzed by NIH Image J software (NIH Online tool-https://ij.imjoy.io/) as we described earlier [32].

### 2.8. Cytokine Assay

NHBE ALI cultures were exposed to CS and infected with HIV. Sixteen hours post infection, the cells were washed apically and basolaterally with PBS four times to remove any residual input virus. The experiments were terminated after 48 h, culture supernatants were collected and proinflammatory cytokines levels were determined using the respective ELISA kits [IL-8/CXCL8 (R & D systems, MN, USA, Cat# D8000C), MCP-1/CCL2 (R & D systems, Cat# DCP00), G-CSF (R & D systems, Cat# DCS50), and CXCL5/ENA-78 (ThermoFisher Scientific, Cat# DY254-05) according to manufacturer’s instructions. Synergy HT Multi-Mode Microplate Reader, from BioTek, Winooski, VT, USA, recorded the color absorbance

### 2.9. Proteomics Profiling for Proinflammatory Cytokines

NHBE ALI cultures were exposed to CS and infected with 5 ng p24 equivalent of HIV BaL (R5- tropic strain). Sixteen hours post-infection, the cells were washed apically and basolaterally with PBS four times to remove any residual input virus, and the cells were then exposed to CS in fresh media. The experiments were terminated after 48 h, and the cells were lysed with RIPA buffer containing a protease inhibitor cocktail, and protein expression was quantified by a proteomics analysis. The proteomics analysis was done using a RayBio Label-Based (L-Series), Human Antibody Array L-8000 Glass Slide Kit, and a combination of Human L-507, L-493, L-3, L-4, L-5, L-6, L-7, L-8, L-9, L-10, L-11, L-12, L-13, L-14, L-15 and L-16 arrays (RayBiotech, GA, USA) was analyzed by TBtools-II, v2.019. Blue indicates a lower intensity of protein expression, and red indicates a higher intensity of protein expression. The color indicates log2 fold-change between uninfected or air-exposed control and HIV-infected/CS-exposed cells.

### 2.10. Statistical Analysis

Unless otherwise mentioned, data were expressed as mean ± SEM from NHBE ALI cultures from lungs from at least three different donors. Data were analyzed using GraphPad Prism software (version 9.5.1). Statistical analysis using unpaired *t*-tests for two groups or ANOVA was followed by Tukey–Kramer honestly significant difference test for multiple comparisons as appropriate. The significance was considered at the level of *p* < 0.05. * = Significant from control; S = significant from each other.

## 3. Results

### 3.1. HIV Tat and CS Alters the Bronchial Epithelial microRNAome and Suppresses SIRT1

NHBE ALI cultures grown on transwells were infected apically and basolaterally with the R5-tropic strain of HIV. Another set was treated with Tat or heat-inactivated Tat as a control to understand the specific role of the Tat protein. In both cases there was a significant upregulation of miR-34a-5p expression (Figure 1A,B).

Analysis of the miRDB and miRTarBase databases identified *SIRT1* as a potential target of miR-34a-5p. Two matching positions for miR-34a-5p binding within the 3′-UTR of *SIRT1* have been reported [33]. To examine their relevance, we evaluated the expression of *SIRT1* in BEAS-2B cells transfected with a miR-34a-5p mimic. qPCR analysis showed that the *SIRT1* mRNA and protein levels were significantly decreased in miR-34a-5p mimic transfected cells (Figure 1C,D).

To further investigate the effect of HIV Tat and CS on *SIRT1* mRNA expression in NHBE ALI cultures, NHBE cells were treated with HIV Tat or a combination of HIV Tat with aurintricarboxylic acid (ATA), a small molecule inhibitor of miRNA processing [34,35]. Another set of NHBE cells were exposed to CS or a combination of CS plus ATA. Our results demonstrate that HIV Tat and CS significantly suppress *SIRT1* mRNA expression, and this suppression was rescued by ATA treatment (Figure 1E,F). Together, these results indicate that HIV Tat and CS alter the bronchial epithelial microRNA, specifically miR-34a-5p, and suppress the *SIRT1* gene expression level.

### 3.2. HIV Infection and CS Suppress SIRT1 Gene Expression Through TGF-β Signaling

Next, we investigated the effect of HIV infection alone, and in combination with CS exposure, on the expression levels of *SIRT1* in NHBE-ALI cultures. NHBE ALI cells were infected with HIV and exposed to CS for eight days, with fresh growth media added every alternate day. Our results reveal a significant decrease in *SIRT1* mRNA levels in cells infected with HIV, and this decrease was enhanced in the presence of CS (Figure 2A). Interestingly, SIRT1 protein levels were completely abolished by either HIV or HIV in combination with CS (Figure 2B).

Our earlier finding shows that CS enhances the HIV infection of NHBE cells by upregulating TGF-β signaling [11,15,20]. Thus, we determined if TGF-β treatment leads to a similar downregulation of *SIRT1* mRNA and protein levels. Our results show that TGF-β treatment alone is sufficient to suppress *SIRT1* mRNA and protein levels in NHBE ALI cultures (Figure 2C,D).

### 3.3. HIV and CS Upregulates ADAM17 Possibly by SIRT1 Suppression

*SIRT1* can directly and indirectly (via *TIMP*) regulate *ADAM17* [36,37], an activator of Notch signaling [26]. Notch ligands, receptors, and downstream molecules regulate multiple cellular processes, including the balance of secretory and ciliated cell differentiation [38,39,40,41,42,43,44,45]. In chronic airway diseases, Notch dysregulation is associated with goblet cell hyperplasia [46], mucus hypersecretion [47], and decreased ciliary cells [48]. Hence, *SIRT1* suppression can lead to attenuated MCC due to reciprocal *ADAM17* upregulation and activation of Notch signaling.

Since HIV Tat alters the bronchial epithelial microRNAome [28] and increases miR-34a-5p levels, which regulate *SIRT1* expression, we then investigated the effect of *SIRT1* suppression on *ADAM17* mRNA regulation by transfecting NHBE cells with anti-SIRT1 siRNA. After confirming successful silencing of *SIRT1* mRNA (Figure 3A) we were able to show that *SIRT1* silencing increased *ADAM17* mRNA levels (Figure 3B). These data suggest that *SIRT1* suppression may be directly involved in regulating *ADAM17* transcription and demonstrates crosstalk between *SIRT1* and *ADAM17*. Furthermore, we found that exposing NHBE cells to HIV with or without CS exposure also induced a significant upregulation of *ADAM17* mRNA and protein levels in NHBE cells (Figure 3C,D). More importantly, we observed synergistic upregulation in *ADAM17* mRNA and protein levels upon HIV infection in combination with CS exposure (Figure 3C,D).

### 3.4. Effect of HIV and CS-Mediated MCC Dysfunction

Proper MCC relies on the presence of mucus, cilia, and a thin layer of airway surface liquid (ASL) to support ciliary beating. Any abnormalities in these components impairs MCC, which in turn can encourage microbial colonization and chronic inflammation [49,50]. Thus, we then investigated whether HIV and CS-mediated *SIRT1* suppression would translate to decreased ciliation and increased levels of *MUC5AC*, a mucin gene associated with airway mucus hypersecretion. qPCR analysis demonstrates a significant upregulation of *MUC5AC* mRNA expression in NHBE cells upon HIV infection (Figure 4A). Surprisingly, co-exposure to HIV plus CS demonstrated a lower level of *MUC5AC* mRNA expression compared to HIV infection alone, but still higher than controls (Figure 4A). However, using immunofluorescence analysis with HIV alone or in combination with CS exposure demonstrated the same increase in MUC5AC expression (Figure 4B,C). Immunofluorescence analysis was also able to demonstrate a decrease in ciliation in NHBE ALI cultures infected with HIV and/or exposure to CS compared to uninfected controls, suggesting a potential switch in differentiation from ciliated cells to goblet cells in response to HIV and CS exposure (Figure 4D).

These results highlight the impact of HIV infection and CS exposure on bronchial epithelial MCC, which manifests as mucus hypersecretion and ciliary dysfunction. While we have observed a synergistic upregulation of *ADAM17*, a failure to observe synergistic downstream effects on *MUC5AC* could be because enough *ADAM17* is already available with HIV infection to mediate its effects on Notch signaling and goblet cell hyperplasia.

### 3.5. SIRT1-Mediated ADAM17 Upregulation Activates Secretion of Proinflammatory Cytokines

PLWH have an increased incidence of lung inflammation and COPD [6]. A number of reports in the literature have shown that ADAM17 directly releases soluble forms of proinflammatory cytokines by cleaving them from their membrane-bound forms [51]. Additionally, ADAM17 can directly boost proinflammatory signaling by shedding TGF-β, which stimulates the production of inflammatory mediators [52].

To determine if *SIRT1* suppression and the consequent *ADAM17* upregulation leads to an increased proinflammatory response, we measured the proinflammatory cytokines in NHBE ALI cultures infected with HIV BaL (R5- tropic strain) alone or in combination with CS exposure using their respective ELISA kits (according to the manufacturer’s instructions). We observed elevated levels of all cytokines tested in culture media, namely IL-8/CXCL8, MCP-1/CCL2, G-CSF, and CXCL5 (Figure 5A–D) in HIV-infected NHBE ALI cultures with or without CS exposure. However, MCP-1, G-CSF, and CXCL5 cytokine levels were higher compared to uninfected air-exposed controls, but lower than HIV infection alone, mirroring the data we observed with *MUC5AC* mRNA. Finally, we also carried out a whole proinflammatory cytokine profile level using RayBio Label-Based (L-Series), Human Antibody Array L-8000 Glass Slide Kit.

This array allows for the simultaneous measurement of the relative abundance of multiple proteins in the sample. Consistent with our ELISA finding, our data revealed the upregulation of additional proinflammatory cytokines, including IL-1a, IL-2, IL-5, IL-6, IL-7, IL-8, IL-9, MCP-1, RAGE, S100 A8/A9, TNF-a and TNF-b (Figure 5E). This uregulation suggests a proinflammatory response in NHBE ALI upon exposure to both HIV and CS. Furthermore, the combination of HIV infection and CS resulted in a further increase in the expression levels of these cytokines compared to infection alone, suggesting a synergistic effect of HIV infection and CS exposure.

## 4. Discussion

Our study provides new insights into the regulatory role of the *SIRT1*/miR-34a-5p/*ADAM17* axis in HIV-associated lung disease, particularly in impaired mucociliary clearance (MCC) and inflammation. We demonstrate that the combination of HIV or HIV Tat protein exposure, with or without cigarette smoke (CS), leads to the suppression of *SIRT1*. These findings are consistent with the role of *SIRT1* as a critical regulator of inflammation, apoptosis, and cellular homeostasis [53].

Our data aligns with recent findings in other diseases, such as diabetic kidney disease (DKD), where *SIRT1* activation mitigates pathological changes by modulating key downstream targets, including *TIMP3* and *ADAM17* [54]. Another study by Jia et al., 2024, showed that *SIRT1* overexpression inhibited *ADAM17* expression in macrophages [55]. Similar to the renal tubular epithelial cells in hyperglycemic conditions described by Ziyu et al., we observed an imbalance in the *SIRT1*/*ADAM17* pathway in airway epithelial cells in the context of HIV and CS exposure. The airway epithelium is the primary barrier against inhaled particles, including allergens, toxicants such as CS, and pathogens. Exposure to CS has been linked to various pathophysiological mechanisms in the airway epithelium, including oxidative stress, activation of CYP enzymes, suppression of *CFTR* biogenesis, mitophagy, and impairment of MCC in normal bronchial epithelial cells [56]. In addition, our laboratory has demonstrated that both X4-tropic and R5-tropic HIV-1 virus, in combination with CS, dysregulated the *CFTR* function which is required to provide optimal ASL depth and efficient MCC [11]. An inadequate level of the CFTR exacerbates the decrease in ASL depth, the impairment of the MCC, and epithelial cellular homeostasis. In addition, several prior studies have identified irregularities in the nasal MCC apparatus in HIV-infected patients [57,58]. Although the pathophysiology underlying lung comorbidities is a topic of continued research, a clear understanding of the pathways involved remains elusive.

HIV-infected cells secrete Tat protein, which can enter the non-infected cells and interrupt many of the host’s immune functions by activating several genes regulated by specific viral and endogenous cellular promoters. In this study, we focused on the *SIRT1* gene, as it plays an important role in HIV-1 transcription and is a potential therapeutic target in airway diseases. Limited research is available on the importance of *SIRT1* in lung inflammation, and to our knowledge, its effect on MCC has not been investigated. Therefore, we focused on the effect of *SIRT1* on the impaired MCC regulated by HIV alone or in combination with CS, and the role of microRNA in the process. Our results demonstrate that the combination of HIV or HIV Tat protein with or without CS leads to a decrease in *SIRT1* levels. *SIRT1* suppression is mediated, at least in part, by the upregulation of miR-34a-5p, which targets *SIRT1*. *SIRT1* is known to regulate cellular processes such as inflammation, apoptosis, and metabolism, and its downregulation can profoundly affect cell function [53].

Our previously published data show that TGF-β signaling plays a prominent role in HIV viral replication and HIV latency, and this could be exacerbated in CS exposure [15]. Our data support prior work that has shown that *ADAM17* upregulation in primary bronchial fibroblast and bronchial biopsies from chronic obstructive asthma patients accelerates subepithelial fibrosis by enhancing extra cellular matrix production and fibroblast differentiation [59]. Moreover, TGF-β signaling has been shown to be involved in *ADAM17* upregulation and increased airway fibrosis [59]. We also demonstrated that *SIRT1* suppression increases *ADAM17* levels in NHBE ALI cultures infected with HIV alone or in combination with CS. We also confirmed that *ADAM17* upregulation is regulated by *SIRT1*, since *SIRT1* siRNA also led to an increase in *ADAM17* levels.

*ADAM17* regulates Notch signaling, which plays a vital role in airway pathologies and has been shown to promote goblet cell hyperplasia with decreased ciliated cells [60]. Our data confirms this by showing decreased ciliated cells and increased *MUC5AC* expressing cells in NHBE ALI cultures infected with HIV with and without CS exposure. This switch in differentiation can impair the ability of the airway epithelium to clear mucus and particulate matter, leading to airway obstruction and exacerbation of respiratory symptoms. Impaired MCC can also lead to the increased incidence of lung infection observed in PLWH. *ADAM17* upregulation can also manifest as increased inflammation, as it is involved in processing and shedding various cell surface proteins, including cytokines like TNF-α and growth factors [51]. This may explain the increased levels of pro-inflammatory cytokines observed in NHBE ALI culture models in our study. Our prior studies demonstrated that HIV and CS suppress *CFTR*, individually or synergistically through TGF-β1 signaling. *CFTR* plays a significant role in MCC, and the current study also confirmed the increased expression of *MUC5AC* with elevated inflammation [11,15,28,35]. The current study builds on these findings by demonstrating increased *MUC5AC* expression alongside elevated inflammation in NHBE ALI cultures exposed to HIV and CS. The overproduction of MUC5AC, a major airway mucin, combined with decreased CFTR expression, likely further impairs MCC, leading to mucus retention and increased susceptibility to lung infections.

In summary, our finding highlights the complex interplay between the altered microRNAome, *SIRT1*, and *ADAM17*, and its downstream effects on MCC and inflammation, and explains the underlying pathophysiology in the increased incidence of lung inflammation, obstructive lung diseases, and lung infections in PLWH-associated airway diseases. These results underscore the importance of addressing HIV reservoirs in the lung as factors in the management of non-AIDS lung comorbidities in PLWH. Despite these findings, knowledge gaps remain regarding the precise molecular mechanism through which the *SIRT1*/*ADAM17* axis works to exacerbate MCC impairment and inflammation in the context of HIV and CS exposure. Strategies to restore *SIRT1* levels or directly modulate the *ADAM17* pathway using gene specific microRNA antagonism may represent promising therapeutic avenues for HIV-associated COPD.

## Figures and Tables

**Figure 1 cells-13-02009-f001:**
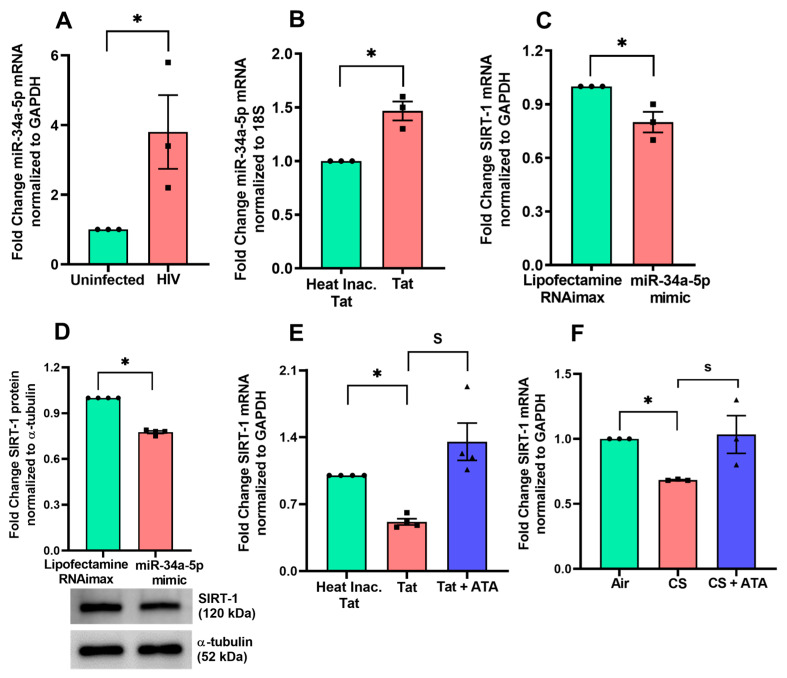
HIV Tat and CS alter the bronchial epithelial microRNAome and suppress SIRT1. (**A**) Primary NHBE cells were infected with five ng p24 equivalent of HIV BaL (R5-tropic strain). Experiments were terminated after 48 h, and total RNA was analyzed. HIV alters the bronchial epithelial microRNAome and upregulates miR-34a-5p. (**B**) Primary NHBE ALI cultures were treated with Tat (10 nM) (heat-inactivated Tat as control). HIV Tat upregulates miR-34a-5p, which is known to suppress *SIRT1*. Even though Tat induces miR-34a-5p by a log2 fold of 8, miR-34a-5p binds to and regulates over 600 different genes with one or more binding sites each (https://awi.cuhk.edu.cn/~miRTarBase/miRTarBase_2025/php/index.php) (accessed on 24 January 2024). Hence, the observed magnitude of *SIRT1* suppression aligns with expectations that an effective increase of miR-34a-5p per target will be much smaller. (**C**) BEAS-2B airway epithelial cells were transfected with miR-34a-5p mimic (40 nM) using RNAiMAX Lipofectamine reagent. The total RNA was extracted, analyzed by qPCR and the *SIRT1* suppression was observed. (**D**) Western blot analysis confirmed the suppression of SIRT1 protein level after transfecting miR-34a-5p mimic. (**E**) NHBE ALI cultures were treated with HIV Tat (heat inactivated Tat as control). Separately, another set was treated with ATA. HIV Tat suppresses *SIRT1* mRNA compared to control, and ATA rescues the level of *SIRT1*. (**F**): NHBE ALI cultures were exposed to CS (air as control). Separately, another set was treated with ATA. CS suppresses *SIRT1* mRNA compared to control, and ATA rescues this. n = NHBE ALI cultures from at least three different lungs, n = 3 different experiments using at least BEAS-2B cells, * = significant from control (*p* < 0.05), S = significant from each other (*p* < 0.05).

**Figure 2 cells-13-02009-f002:**
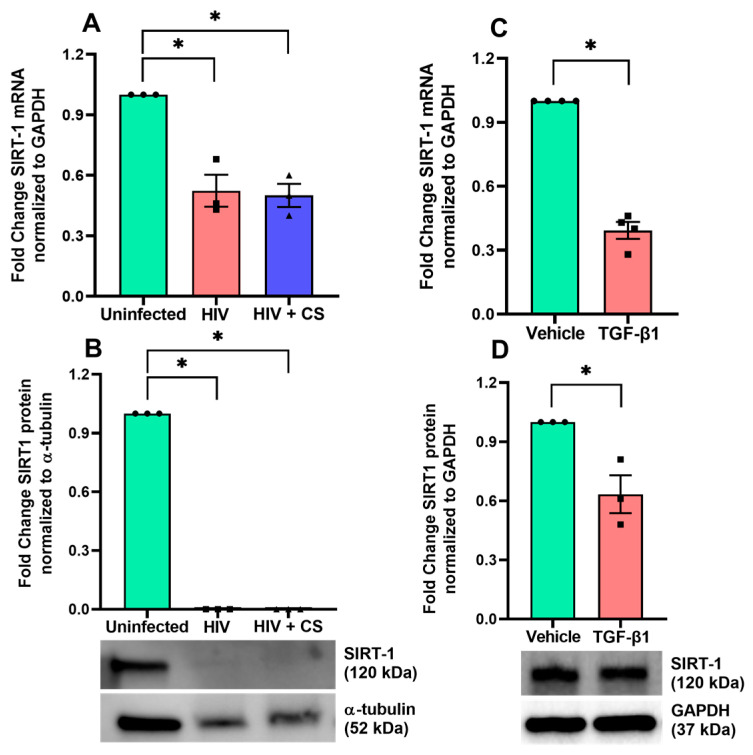
HIV infection and CS suppress *SIRT1* gene expression through TGF-β signaling. (**A**) NHBE ALI cultures were exposed to CS and infected with five ng p24 equivalent of HIV BaL (R5- tropic strain). Experiments were terminated after 48 h, and total RNA was analyzed for *SIRT1* mRNA by qPCR. The combination of HIV and CS exposure suppresses *SIRT1* expression compared to uninfected. (**B**) Similar set of experiments were used to analyze for SIRT1 protein level by Western blot. HIV infection alone and combination of HIV and CS exposure suppress SIRT1 protein level. (**C**) NHBE ALI cultures were treated with recombinant TGF-β (10 ng/mL, vehicle as control). Forty-eight hours post-treatment, total RNA was isolated and analyzed for *SIRT1* mRNA expression level using qPCR. The *SIRT1* mRNA level was significantly lower in TGF-β-treated NHBE cells than in the vehicle-treated control. (**D**) Forty-eight hours post-treatment with TGF-β1, total protein was isolated and analyzed for SIRT1 protein expression level using western blot. The SIRT1 protein level was significantly lower in TGF-β1 treated NHBE cells than in the vehicle control. n = NHBE ALI cultures from at least three different lungs, * = significant from control (*p* < 0.05).

**Figure 3 cells-13-02009-f003:**
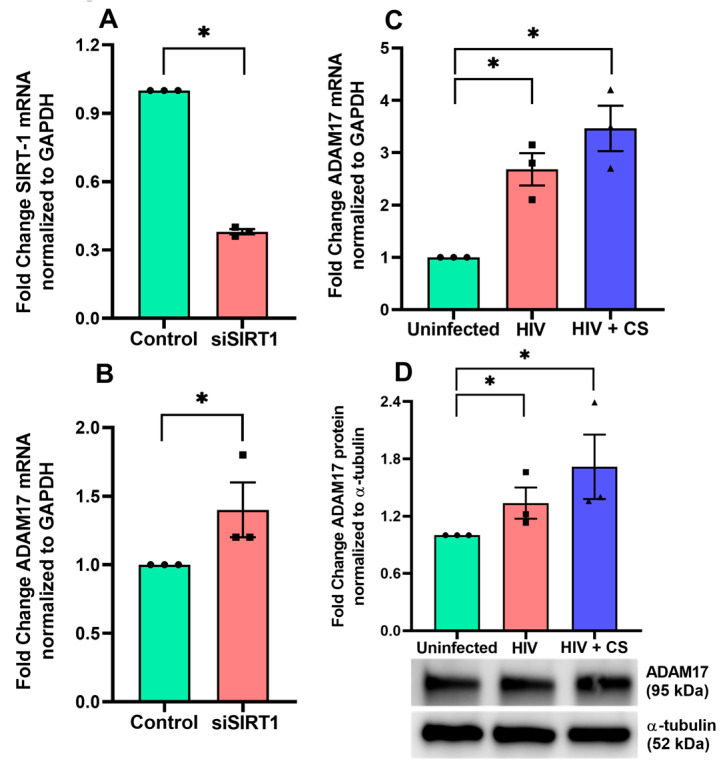
HIV and CS upregulate *ADAM17* possibly by *SIRT1* suppression. (**A**) To investigate the effect of *SIRT1* on *ADAM17* regulation, NHBE cells were transfected with SIRT1-siRNA using RNAiMAX Lipofectamine transfection reagent. Total RNA was isolated and analyzed for gene expression. qPCR analysis confirmed the successful silencing of *SIRT1*. (**B**) qPCR analysis confirmed the upregulation of *ADAM 17* upon *SIRT1* silencing. (**C**) NHBE ALI cultures were exposed to CS and infected with five ng p24 equivalent of HIV BaL (R5-tropic strain). Experiments were terminated after 48 h, and total RNA was analyzed for *ADAM17* mRNA by qPCR. HIV-infected and the combination of HIV plus CS exposure increases the *ADAM17* mRNA level. (**D**) Another experimental set was used for ADAM17 protein level by western blot. The upregulation of ADAM17 protein levels were noticed in both HIV and CS-exposed NHBE ALI cultures. n = NHBE ALI cultures from at least 3 different lungs, * = significant from control (*p* < 0.05).

**Figure 4 cells-13-02009-f004:**
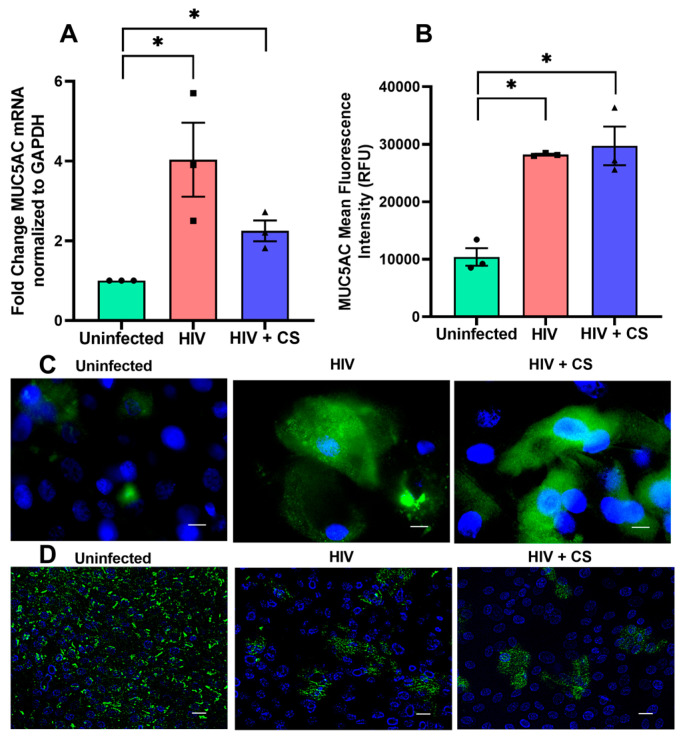
Effect of HIV and CS-mediated MCC dysfunction. (**A**) NHBE ALI cultures were exposed to CS and infected with five ng p24 equivalent of HIV BaL (R5- tropic strain). Experiments were terminated after 9 days, and total RNA was analyzed for *MUC5AC* mRNA by qPCR. HIV and CS increase the *MUC5AC* mRNA level compared to uninfected. (**B**,**C**) MUC5AC immunofluorescent staining of NHBE ALI cultures and protocol details described in method section. MUC5AC in green and nuclei stained with DAPI in blue. Upregulation of MUC5AC was observed upon HIV infection and in CS-exposed cells compared to uninfected, scale-10 µM. (**D**) HIV and CS inhibit ciliogenesis. Primary bronchial epithelial cells were exposed to CS and infected with HIV. On Day 9 post-infection, ciliogenesis was observed in lung-matched uninfected control, and protocol details described in method section. Cilia in green; nuclei stained with DAPI in blue. We observed impaired ciliation in both HIV-infected and CS-exposed NHBE ALI cultures, scale-20 µM. n = NHBE ALI cultures from 3 different lungs at least, * = significant from control (*p* < 0.05).

**Figure 5 cells-13-02009-f005:**
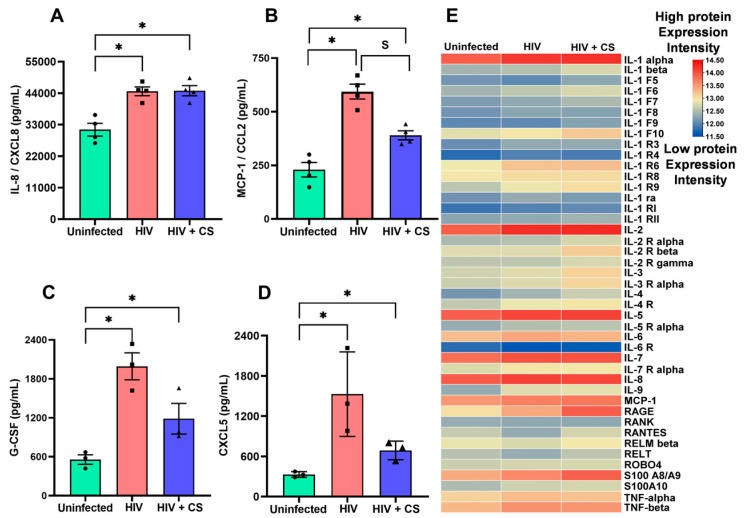
*SIRT1*-mediated *ADAM17* upregulation activates the cytokine storm. (**A**–**D**) NHBE ALI cultures were exposed to CS and infected with five ng p24 equivalent of HIV BaL (R5-tropic strain). Experiments were terminated after 48 h, and the supernatant was analyzed for quantitative measurement of secretory cytokines IL-8/CXCL8 and MCP-1/CCL2, G-CSF and CXCL5. ELISA results demonstrated a significant upregulation of IL-8, CCL2, G-CSF, and CXCL5 in NHBE cells upon infection compared to uninfected controls. (**E**) Experiments were terminated after 48 h, and total protein was analyzed for proinflammatory cytokine protein levels using RayBio Label-Based (L-Series), Human Antibody Array L-8000 Glass Slide Kit. Heatmap analysis shows the upregulated proinflammatory cytokine protein level, including IL-1α, IL-2, IL-5, IL-6, IL-7, IL-8, IL-9, MCP-1, RAGE, S100 A8/A9, TNF-α and TNF-β. n = NHBE ALI cultures from at least three different lungs, * = significant from control (*p* < 0.05), s = significant between the treatment group.

## Data Availability

All data generated or analyzed during this study are included in this published article.

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
