# Peer review of "HIV-1 Tat Protein and Cigarette Smoke Mediated ADAM17 Upregulation Can Lead to Impaired Mucociliary Clearance"

_cells, 2024, doi:10.3390/cells13232009_

Round 1

Reviewer 1 Report

Comments and Suggestions for Authors

The authors examined the effects of HIV infection and cigarette smoking (CS) on SIRT1, ADAM17, mucociliary clearance (MCC), and inflammatory cytokines in normal human bronchial epithelial (NHBE) cells. The major findings are: 1) HIV and tat increased miR-34a-5p; a miR-34a-5p mimic suppressed SIRT1; tat and CS suppressed SIRT1 mRNA. 2) HIV and CS suppressed SIRT1 mRNA and protein levels, so did TGF-beta1. 3) siSIRT1 upregulated ADAM17 mRNA; HIV and HIV+CS upregulated ADAM17 mRNA and protein levels. 4) HIV and HIV+CS upregulated MUC5AC mRNA and protein, which is associated with airway mucus hypersecretion, thus mediating MCC dysfunction. 5) HIV and HIV+CS induced proinflammatory cytokines. The authors concluded that HIV infection (probably tat) and CS impair bronchial MCC through suppression of SIRT1, leading to ADAM17 upregulation, which activates Notch signaling and induces cytokine storm to impair MCC. The data supported the major findings and conclusions, although the analyses were limited to the NHBE culture system.

A few specifics are missing:

Where was the source for HIV tat?  How was tat heat-inactivated?

Where was the source for TGF-beta1?

Other than Figure 1, HIV tat was not tested for the experiments in other figures.  Was there a reason for not testing tat?  Please comment.

Ref 56 Tilley et al, Am J Respir Crit Care Med 2009, reported "Down-regulation of the notch pathway in human airway epithelium in association with smoking and chronic obstructive pulmonary disease", which is in the opposite direction of ADAM17 upregulation by CS.  What would be the explanations for the contradictory findings?

Lines 510-515 appear to contain text errors that need to be fixed.

Author Response

Reviewer 1

Yes

Can be improved

Must be improved

Not applicable

Does the introduction provide sufficient background and include all relevant references?

(x)

( )

( )

( )

Is the research design appropriate?

(x)

( )

( )

( )

Are the methods adequately described?

( )

(x)

( )

( )

Are the results clearly presented?

(x)

( )

( )

( )

Are the conclusions supported by the results?

(x)

( )

( )

( )

The authors examined the effects of HIV infection and cigarette smoking (CS) on SIRT1, ADAM17, mucociliary clearance (MCC), and inflammatory cytokines in normal human bronchial epithelial (NHBE) cells. The major findings are: 1) HIV and tat increased miR-34a-5p; a miR-34a-5p mimic suppressed SIRT1; tat and CS suppressed SIRT1 mRNA. 2) HIV and CS suppressed SIRT1 mRNA and protein levels, so did TGF-beta1. 3) siSIRT1 upregulated ADAM17 mRNA; HIV and HIV+CS upregulated ADAM17 mRNA and protein levels. 4) HIV and HIV+CS upregulated MUC5AC mRNA and protein, which is associated with airway mucus hypersecretion, thus mediating MCC dysfunction. 5) HIV and HIV+CS induced proinflammatory cytokines. The authors concluded that HIV infection (probably tat) and CS impair bronchial MCC through suppression of SIRT1, leading to ADAM17 upregulation, which activates Notch signaling and induces cytokine storm to impair MCC. The data supported the major findings and conclusions, although the analyses were limited to the NHBE culture system.

A few specifics are missing:

Where was the source for HIV tat?  How was tat heat-inactivated?

Reply: We have added the information in the revised version. Changes done on – Line no. 134-136, Pg No. 3.

Where was the source for TGF-beta1?

Reply: We have added the information in the revised version. Changes done on – Line no. 136-138, Pg No. 3.

Other than Figure 1, HIV tat was not tested for the experiments in other figures.  Was there a reason for not testing tat?  Please comment.

Reply: We used HIV Tat specifically in Figure 1 to demonstrate the microRNA-mediated suppression of SIRT1. In the other experiments, we focused on the broader context of HIV infection and cigarette smoke exposure to investigate their combined effects on the microRNA mediated gene suppression.

Ref 56 Tilley et al, Am J Respir Crit Care Med 2009, reported "Down-regulation of the notch pathway in human airway epithelium in association with smoking and chronic obstructive pulmonary disease", which is in the opposite direction of ADAM17 upregulation by CS.  What would be the explanations for the contradictory findings?

Reply: Thank you for your suggestions. We have cross-checked with the mentioned article regarding the expression of ADAM17 in patients with COPD. The authors have not provided any information on ADAM17 gene expression, either down-regulation or up-regulation.

Lines 510-515 appear to contain text errors that need to be fixed.

Reply: We have corrected in the revised version. Changes done on – Line no. 530-541, Pg No. 13.

Reviewer 2 Report

Comments and Suggestions for Authors

The manuscript submitted by Kingshuk Panda et al. explores the impact of HIV-1 Tat protein and smoking on airway epithelial cells, particularly in terms of mucociliary clearance and inflammatory responses. The study provides valuable insights and may be of significant importance for understanding pulmonary complications in HIV-infected individuals. However, to enhance the quality of the manuscript, the authors are advised to consider the following suggestions, especially regarding the presentation of results and the depth of discussion.

1. It is recommended that the authors check whether the title and keywords accurately reflect the core content and innovative aspects of the research. For instance, if "HIV-1 Tat protein" and "cigarette smoke" are key factors in the study, ensure that these terms are prominently featured in the title.

2. It is suggested that the authors expand the introduction section to discuss in more detail the progress of research on the impact of HIV-1 and smoking on pulmonary complications, as well as how this study fills the gaps in existing knowledge.

3. It is advised that the authors provide more details in the Materials and Methods section.

4. The authors are encouraged to provide more details in the statistical analysis section, including the statistical tests used, the threshold for P-values, and whether the data meet the assumptions of these tests.

5. There are some garbled elements in the manuscript, particularly the line numbers.

6. It is recommended that the authors delve deeper into the significance of the study's results in the discussion section, including how they relate to existing literature and their potential impact on clinical practice.

7. The authors are advised to check the currency of the references and ensure that all relevant and up-to-date studies are cited to support their arguments and conclusions. Additionally, it may be worth considering whether the number of references could be streamlined.

Author Response

Reviewer 2

Yes

Can be improved

Must be improved

Not applicable

Does the introduction provide sufficient background and include all relevant references?

( )

(x)

( )

( )

Is the research design appropriate?

( )

(x)

( )

( )

Are the methods adequately described?

( )

(x)

( )

( )

Are the results clearly presented?

( )

(x)

( )

( )

Are the conclusions supported by the results?

( )

(x)

( )

( )

The manuscript submitted by Kingshuk Panda et al. explores the impact of HIV-1 Tat protein and smoking on airway epithelial cells, particularly in terms of mucociliary clearance and inflammatory responses. The study provides valuable insights and may be of significant importance for understanding pulmonary complications in HIV-infected individuals. However, to enhance the quality of the manuscript, the authors are advised to consider the following suggestions, especially regarding the presentation of results and the depth of discussion.

  1. It is recommended that the authors check whether the title and keywords accurately reflect the core content and innovative aspects of the research. For instance, if "HIV-1 Tat protein" and "cigarette smoke" are key factors in the study, ensure that these terms are prominently featured in the title.

Reply: Thank you for your suggestion. Our tittle already has both the term “HIV-1 Tat protein" and "cigarette smoke".

  1. It is suggested that the authors expand the introduction section to discuss in more detail the progress of research on the impact of HIV-1 and smoking on pulmonary complications, as well as how this study fills the gaps in existing knowledge.

Reply: Thank you for your suggestion. We have added in the revised version. Changes done on – Line no. 65-67, Pg No. 2

  1. It is advised that the authors provide more details in the Materials and Methods section.

Reply: As per the reviewer suggested, we have added more details in the material and methods section. Changes done on – Line no. 134-138, Pg No. 3.

  1. The authors are encouraged to provide more details in the statistical analysis section, including the statistical tests used, the threshold for P-values, and whether the data meets the assumptions of these tests.

Reply: As suggested, we have added more details to the statistical analysis section. Changes done on – Line no. 230, 233; Pg No. 5.

  1. There are some garbled elements in the manuscript, particularly the line numbers.

Reply: We have corrected in the revised version.

  1. It is recommended that the authors delve deeper into the significance of the study's results in the discussion section, including how they relate to existing literature and their potential impact on clinical practice.

Reply: Thank you for the suggestion. We have revised the discussion section to provide a more in-depth analysis of the significance of our findings. Changes done on – Line no. 475-487, Reference no. 56, 57, 58, Pg No. 11-12

  1. The authors are advised to check the currency of the references and ensure that all relevant and up-to-date studies are cited to support their arguments and conclusions. Additionally, it may be worth considering whether the number of references could be streamlined.

Reply: Thank you for your suggestion. We have reviewed and included the references to ensure they are current and relevant. Changes done on – Reference no. 56, 57, 58.

Reviewer 3 Report

Comments and Suggestions for Authors

The study addresses crucial findings in HIV infection and COPD highlighting the importance of the SIRT1/ADAM17 pathway in epithelial dysfunction. 

1. Please elaborate on your discussion section, can you summarize your findings in para 1 and then co-relate with the literature? 

2.  All experiments are in vitro, have you done any experiments in vivo? or do you plan to work on a mice model? Your experiments suggest having a smoking robot, can that be used in vivo?

3. have you tried mRNA studies using lung tissues from smokers' and non-smokers' human lungs?

4. Your study completely relies on one culture model, HBEcs, have you tried it with other cell types (HPAECs or HLMVECs)

5. Can you discuss the potential limitations of your study? ex: clinical translation and challenges 

6. Please address the knowledge gaps in your discussions, ex: align your findings with known COPD pathways or mechanisms

7. What are the future directions of your study?

8. Elaborately discuss the pathway SIRT1/ADAM17 ex: do you have references to which you can relate your findings? 

9. Can you add details on available antiretroviral therapies available? and the potential they have or might have in PLWH.

10. Do you have lighter exposure in Figure 2 D and 3D western blot? though your analysis shows an increase or decrease the representative blots doesn't complement it.  

11. Fig 1 after transfection what % of expression decrease have you seen in SIRT-1? Is it consistent for all experiments? 

12. have you performed any over-expression studies 

Author Response

Reviewer 3

Yes

Can be improved

Must be improved

Not applicable

Does the introduction provide sufficient background and include all relevant references?

(x)

( )

( )

( )

Is the research design appropriate?

(x)

( )

( )

( )

Are the methods adequately described?

(x)

( )

( )

( )

Are the results clearly presented?

(x)

( )

( )

( )

Are the conclusions supported by the results?

( )

(x)

( )

( )

Comments and Suggestions for Authors

The study addresses crucial findings in HIV infection and COPD highlighting the importance of the SIRT1/ADAM17 pathway in epithelial dysfunction. 

  1. Please elaborate on your discussion section, can you summarize your findings in para 1 and then co-relate with the literature? 

Reply: As suggested, we have summarized the findings in para 1 and discussed with literature. Changes done on – Line no. 475-487, Pg No. 11-12.

  1. All experiments are in vitro, have you done any experiments in vivo? or do you plan to work on a mice model? Your experiments suggest having a smoking robot, can that be used in vivo?

Reply: Thank you for your insightful comment. Yes, we plan to extend our work to in-vivo studies using a mouse model, which will be outlined in our next manuscript. This current study represents the first report of these findings in NHBE cells cultured at air-liquid interface (ALI). Our primary focus here was on in vitro experiments to establish baseline mechanisms and validate our hypotheses. Regarding the smoking robot, while it is designed primarily for in vitro studies, its application can potentially be adapted for in vivo smoke exposure studies. We plan to expose mice models with cigarette smoke using this robot. We appreciate your suggestion and will consider it for future research.

  1. Have you tried mRNA studies using lung tissues from smokers' and non-smokers' human lungs?

Reply: At present, we do not have access to lung tissues from smokers or non-smokers for mRNA studies. Our current work primarily focuses on in vitro models to establish key mechanisms. However, we recognize the importance of validating these findings in human lung tissues and plan to include such analyses in future studies as suitable samples become available.

  1. Your study completely relies on one culture model, HBEcs, have you tried it with other cell types (HPAECs or HLMVECs)

Reply: Thank you for your comment. Our study primarily focuses on the 3d model of bronchial epithelial cells present in the Air-liquid interface to mimic the human airway epithelium. In addition to NHBE cells, we have also conducted experiments using BEAS-2B cell line to validate our findings. Utilizing these two distinct airway epithelial cell models strengthens the reliability of our results. We haven’t performed any experiments on HPAECs or HLMVECs cells.

  1. Can you discuss the potential limitations of your study? ex: clinical translation and challenges

Reply: We acknowledge that our study has certain limitations, particularly regarding clinical translation. While our findings provide significant mechanistic insights using in vitro models, further validation in vivo systems are necessary. In future studies, we plan to work with in vivo models of cigarette smoke (CS) exposure and HIV infection to better mimic the clinical context and strengthen the translational relevance of our work.

  1. Please address the knowledge gaps in your discussions, ex: align your findings with known COPD pathways or mechanisms

Reply: Thank you for addressing this. We have included the statement in the revised version. Changes done on – Line no. 547-552, Pg No. 13.

  1. What are the future directions of your study?

Reply: Future directions of our study include exploring strategies to restore SIRT1 levels or directly modulate the ADAM17 pathway as potential therapeutic approaches for HIV-associated COPD. Additionally, we aim to investigate these mechanisms further in vivo models to validate their therapeutic potential and assess the feasibility of clinical translation. We have added the statement in the revised version. Line no. 550-552, Pg no. 13.

  1. Elaborately discuss the pathway SIRT1/ADAM17 ex: do you have references to which you can relate your findings? 

Reply: Thank you for the suggestion. We have included in the revised version. Changes done on – Line no. 475-487, Pg No.12

  1. Can you add details on available antiretroviral therapies available? and the potential they have or might have in PLWH.

Reply: Thank you for the suggestions. We have added the statement to the revised version. Changes done on – Line no. 41-46, Pg No. 1

  1. Do you have lighter exposure in Figure 2 D and 3D western blot? though your analysis shows an increase or decrease the representative blots doesn't complement it.  

Reply: Thank you for your observation. We used minimum setting 1 sec for the blot images in Figures 2D and 3D. We used densitometric analysis method for quantification without any bias. While the quantification reflects the observed changes accurately, we acknowledge that the representative blots may appear less distinct and provide the raw image as a supplement figure.

  1. Fig 1 after transfection what % of expression decrease have you seen in SIRT-1? Is it consistent for all experiments? 

Reply: Thank you for your question. After transfection, we observed a decrease of approximately 35% in SIRT1 expression, as determined by qRT-PCR or Western blot. This decrease was consistent across all experiments, with minimal variability.

  1. have you performed any over-expression studies 

Reply: Thank you for your suggestion. No, we have not performed over-expression studies in the current work. However, this is an important avenue for further exploration, and I plan to incorporate such studies in future research to provide additional insights.
